

# Disentangling timing and amplitude errors in streamflow simulations

Simon Paul Seibert[1], Uwe Ehret[1], and Erwin Zehe[1]

[1]Karlsruhe Institute of Technology (KIT), Institute for Water and River Basin Management, Chair of Hydrology, Kaiserstrasse 12, 76131 Karlsruhe Germany

*Correspondence to:* Uwe Ehret (uwe.ehret@kit.edu)

**Abstract.** This article introduces an improvement in the Series Distance ($SD$) approach for improved discrimination and visualisation of timing and magnitude uncertainties in streamflow simulations. $SD$ emulates visual hydrograph comparison by distinguishing periods of low-flow and periods of rise and recession in hydrological events. Within these periods, it determines the distance

of two hydrographs not between points of equal time, but between points that are hydrologically similar. The improvement comprises an automated procedure to emulate visual "coarse-graining", i.e. the determination of an optimal level of generalization when comparing two hydrographs, a scaled error model which is better applicable across large discharge ranges than its non-scaled counterpart, and "error dressing", a concept to construct uncertainty ranges around deterministic simulations or

forecasts. Error dressing includes an approach to sample empirical error distributions by increasing variance contribution, which can be extended from standard 1-dimensional distributions to the 2-dimensional distributions of combined time and magnitude errors provided by $SD$.

In a case study we apply both the $SD$ concept and a benchmark model ($BM$) based on standard magnitude errors to a six-year time series of observations and simulations from a small alpine catch-

ment. Time-magnitude error characteristics for low-flow, rising and falling limbs of events were substantially different. Their separate treatment within $SD$ therefore preserves useful information which can be used for differentiated model diagnostics, and which is not contained in standard criteria like the Nash-Sutcliffe-Efficiency. Construction of uncertainty ranges based on the magnitude of errors of the $BM$ approach and the combined time- and magnitude errors of the $SD$ approach

revealed that the $BM$ derived ranges were visually more narrow and statistically superior to the $SD$ ranges. This suggests that the combined use of time- and magnitude errors to construct uncertainty envelopes implies a trade-off between the added value of explicitly considering timing errors and the associated, inevitable time-spreading effect which "inflates" the related uncertainty ranges. Which effect dominates depends on the characteristics of timing errors in the hydrographs at hand. Our

findings corroborate that Series Distance is an elaborated concept for the comparison of simulated and observed stream flow time series which can be used for detailed hydrological analysis, model diagnostics and to inform about uncertainties related to hydrological predictions.





## 1 Introduction

Manifold epistemic and aleatory uncertainties make the simulation of streamflow a fairly uncertain task. Assessment of uncertainties, i.e. quantification, evaluation and communication is thus of great concern in decision making, model evaluation, the design of technical structures like flood protection dams or weirs and many other issues. Every quantification and evaluation of uncertainties involves the comparison of simulated and observed rainfall runoff response.

For this purpose, visual hydrograph inspection is still the most widely used technique in Hydrology as it allows for the simultaneous consideration of various aspects such as the occurrence of hydrological (rainfall-runoff) events, the timing of peaks and troughs, the agreement in shape and the comparison of individual rising or falling limbs within an event. The main strength of visual hydrograph comparison results from the human ability to identify and compare matching, i.e. hydrologically similar parts of hydrographs ("to compare apples with apples") and particularly to discriminate vertical (magnitude) and horizontal (timing) agreement of hydrographs. Whereas the former implies that rising and falling limbs of the two time series are intuitively and meaningfully matched before they are compared, the latter refers to a joint but yet individual consideration of timing and magnitude errors. Visual hydrograph inspection is hence a powerful yet demanding evaluation technique which is still rather difficult to mimic by automated methods. Clear disadvantages of visual hydrograph inspection however are its subjectivity and that its application is restricted to a limited number of events.

### 1.1 Single and multiple criteria for hydrograph evaluation

To overcome this shortcoming, a large number of numerical criteria (Nash and Sutcliffe, 1970; Legates and McCabe, 1999; Pachepsky et al., 2006; Dawson et al., 2007; Laio and Tamea, 2007; Bennett et al., 2013) have been proposed. However, each criterion typically evaluates only one or just a few hydrograph aspects and there is no "one size fits all" solution available. For this reason different attempts have been undertaken to compare expert judgement and automated criteria (Crochemore et al., 2014) and to establish model evaluation guidelines (e.g., Moriasi et al., 2007; Biondi et al., 2012; Harmel et al., 2014). Key points of related guidelines typically include that the choice of the metric should depend i) on the modelling purpose, ii) on the modelling mode (calibration, validation, simulation or forecast) and iii) the model resolution (time stepping, spatial resolution). Further, most authors recommend the combination of several, preferably orthogonal criteria, which might imply combined application of absolute and relative criteria (Willmott, 1981). Hence, within the last decade several multi-criteria approaches for model calibration and evaluation have been proposed (Gupta et al., 1998; Boyle et al., 2000; Vrugt et al., 2003; Efstratiadis and Koutsoyiannis, 2010; Kollat et al., 2012), which combine different performance criteria and/or evaluation against "hydrological signatures" such as the shape of the flow duration curve (Euser et al., 2013; Hrachowitz et al.,





2014). Even approaches aiming to mimic visual hydrograph comparison were developed. These include multicomponent mapping (Pappenberger and Beven, 2004), self-organizing maps (Reusser et al., 2009), wavelets (Liu et al., 2011), the hydrograph matching algorithm Ewen (2011) and the "Peak-Box" approach for interpretation and verification of operational ensemble peak-flow forecasts (Zappa et al., 2013). Despite this considerable progress, many practical and scientific applications (Haag et al., 2005; Gassmann et al., 2013; Seibert et al., 2014; Wrede et al., 2014; Kelleher et al., 2015; Zhang et al., 2016) still rely on simple "Mean Squared Error" (MSE) type distance metrics such as the long established Nash-Sutcliffe-Efficiency (NASH) or the Root Mean Squared Error (RMSE) even though their shortcomings are well known (Seibert, 2001; Schaefli and Gupta, 2007; Gupta et al., 2009).

A less recognized issue of MSE-type criteria is that these compare points with identical abscissa, i.e. at the same position in time. This means that points in the observation are "vertically" compared to points in the simulation (in the following we refer to them as vertical metrics). The problem with this is that small errors in timing may be expressed as large errors in magnitude. It is obvious that neither individual criteria nor the combination of different vertical metrics within a multi-objective approach can compensate for this.

### 1.2 Uncertainty assessment and model diagnostics - learning from model deficiencies

Just as with performance criteria, many methods related to quantification, visualisation and communication of uncertainties were developed in recent decades, and the value of knowledge about simulation uncertainty is now generally acknowledged. The range of methods is large and comprises manifold probabilistic and non-probabilistic approaches. Probabilistic concepts for instance include the total model uncertainty concept (Montanari and Grossi, 2008), methods based on Bayes' Theorem (Krzysztofowicz, 1999; Krzysztofowicz and Kelly, 2000) and various ensemble techniques (Roulston and Smith, 2003; Georgakakos et al., 2004; Cloke and Pappenberger, 2008). Non-probabilistic methods include the generalized likelihood uncertainty estimation (GLUE) (Beven and Binley, 1992), possibilistic methods (Jacquin and Shamseldin, 2007) or approaches applying Fuzzy-Set theory (Nasseri et al., 2014). Uncertainty assessment is a field of ongoing research and so far there is no generally accepted technique available. The most important points of criticism of the non-probabilistic methods are their subjectivity and their inconsistency with probabilistic approaches when these are applied to cases which can be explicitly answered using statistical approaches (Stedinger et al., 2008). On the other hand, probabilistic approaches always rely on the assumptions of ergodicity and stationarity, which are rarely fulfilled in reality. A spin-off of uncertainty assessment is the field of model diagnostics, which ultimately aims to learn more about and from model deficiencies. Related approaches either analyse the temporal patterns of parameter identifiability (Wagener et al., 2003) or the coincidence of typical errors (Reusser et al., 2009) and parameter sensitivity (Reusser and Zehe, 2011) in stream flow simulation.



Motivated by the limitations of vertical distance metrics, Ehret and Zehe (2011) developed the Se-
ries Distance ($SD$) approach. $SD$ is not a single equation but rather a concept designed for joint but
separated assessment of timing and magnitude errors in stream flow simulations, either for events in
distinct periods or the entire time series. "Joint but separated" means that both the time and magni-
tude distances between the observed and simulated hydrographs are determined for "matching pairs
of points" in the event, but the two distances are kept separate. Such separate treatment is for instance
desirable in flood forecasting, where errors in magnitude are relevant for dike defence, whereas er-
rors in timing are crucial for reservoir operation. The separation of timing and magnitude errors is
further helpful for improving model diagnostics as they point towards different deficiencies in the
model structure.

Here we present substantial improvements (Section 2) to the original approach of Ehret and Zehe
(2011), particularly the coarse-graining procedure. We furthermore introduce a heuristic approach
to visualize timing and magnitude uncertainties in streamflow simulations by constructing two-
dimensional uncertainty ranges in section 3. Related to that, we provide and test several quality
criteria to evaluate deterministic uncertainty ranges. The skill of uncertainty ranges is still rarely
evaluated in Hydrology (Franz and Hogue, 2011) and most of the available methods such as rank
probability scores (Duan et al., 2007), rank histograms or the usage of different moments of the
probability density function (De Lannoy et al., 2006) were developed in climatology (Gneiting et al.,
2008; Franz and Hogue, 2011). These approaches typically quantify ensemble spread and thus are
probabilistic approaches to evaluate uncertainty estimation. To our knowledge only few deterministic
approaches (e.g. categorical statistics such as the Brier score or contingency tables) or combinations
of deterministic and probabilistic approaches (Shrestha et al., 2009) are available. In section 4 we
test the feasibility of the advanced $SD$ approach in a case study and compare it to a standard bench-
mark error model. Chapter 5 contains the results and discussion, chapter 6 the related conclusions.
To foster the use of the $SD$ approach, we publish the $SD$ (Matlab) code (licensed under Creative
Commons "BY-NC-SA 4.0") together with a ready-to-use sample data set alongside this manuscript.
It is accessible via a GitHub repository https://github.com/KIT-HYD/SeriesDistance.

## 2   Series Distance - concept and modifications

Series Distance ($SD$) was developed to resemble the strengths of visual hydrograph inspection in an
automated procedure, which typically rests on the following premises (Ehret and Zehe, 2011):

– Hydrographs contain individual "events" separated by periods of low-flow.

– Events are composed of rising and falling limbs (or segments) which are separated by peaks
  and troughs.



– These different parts of event hydrographs reflect different hydro-meteorological processes and should be compared individually, so as to not compare apples with oranges. This is of particular importance if the simulated (sim in the following) and observed (obs in the following) hydrograph do at the same time step $t$ belong to different parts of the hydrograph (compare black rectangle in Fig. 1).

– A comprehensive evaluation of the agreement of matching rising and falling limbs of two hydrographs requires consideration of both errors in timing and magnitude as this better informs us about ways to improve the model. A simulated rising limb can for example match perfectly with its observed counterpart with respect to values, but occur systematically too early (or too late), which would indicate the need to adjust model parameters related to runoff concentration and flood routing or to improve the related model components.

– A comprehensive comparison of sim and obs should also provide information on the overall agreement with respect to the occurrence of relevant events and times of low-flow. This is typically expressed by contingency tables and contains information about correctly predicted, missed and falsely predicted events.

These criteria listed above inform about different error sources and their individual evaluation therefore provides useful information for a targeted model improvement. As $SD$ accounts for all of these aspects it is not a single formula but rather a procedure which includes the following steps. For each step, the main innovations are described in detail in the sections below.

– Hydrograph preprocessing (chapter 2.1). New: Routines to create gap-free, non-negative time series and to filter irrelevant fluctuations.

– Identification and pairing of events (chapter 2.2). New: Routines to read user-specified events and to treat the entire time series as a single, long event.

– Identification, matching and coarse-graining of segments (chapter 2.3): New: This part has been completely reworked and now applies the coarse-graining procedure.

– Calculation of the distance between matching segments with respect to both timing and magnitude (chapter 2.4). This is the core of $SD$, and it is important to note that the distances are computed between points of the hydrographs considered to be hydrologically similar. New: Routines to calculate a scaled magnitude error.

– Calculation of a contingency table which counts matching, missing and false events. No changes.

– FIGURE 1: SD Concept –





### 2.1 Hydrograph preprocessing

Application of $SD$ usually requires some pre-processing to assure gap-free and non-negative time series of equal length; related routines are now included in the $SD$ code. Further routines are available for the adjustment of consecutive identical values (identification of rising and falling limbs requires non-zero gradients) and for time series smoothing (observed time series often contain sensor-related micro-fluctuations which create many non-relevant micro-segments). Smoothing is based on the

Douglas-Peucker algorithm (Douglas and Peucker, 1973) which preserves extremes but filters the noise (Ehret, 2016). Preprocessing also involves the identification of "segments", i.e. contiguous periods of rise or fall in the hydrograph. This is based on the slope of the hydrograph computed between two successive time steps.

### 2.2 Identification and pairing of events

For many aspects of Hydrology such as flood forecasting or studies of rainfall-runoff transformation, it is useful to consider a hydrograph as a succession of distinct events (usually triggered by rainfall events) separated by periods of low-flow. As $SD$ is based on the concept of comparing similar parts of obs and sim hydrographs, it ideally also involves the steps of identifying events both in the obs and sim time series, and then relating the resulting events between the series. On this level, the general

agreement of the two series is evaluated with a contingency table which counts the number of hits (observed events that have a matching simulated counterpart), misses (observed events without a simulated counterpart) and false alarms (simulated events without an observed counterpart). This is also the basis for the further steps of the $SD$ procedure: Only for matching pairs of obs-sim events, matching segments of rise and fall within the events can be identified and the combined

time-magnitude error be computed. For misses, false alarms and periods of low-flow this is not possible. For these cases, the best indicator for hydrological similarity in obs and sim is similarity in time, i.e. the distance between the observed and simulated hydrograph can be computed with a standard vertical distance measure. The detection of events in hydrographs and their subsequent pairing however is not trivial and has to our knowledge not yet been solved in an automated and

generalized way. The original version of $SD$ applied a simple "no-event threshold" (see Fig. 1) which, however, often produced unsatisfactory results in the form of many non-intuitive misses or false alarms if the events peaked just above or below the threshold. To overcome these limitations, two further options are now included in $SD$: The first allows reading of event start and end points and matching obs and sim events from user-provided lists. This option allows users to apply any

desired event detection method such as those proposed by Blume et al. (2007); Seibert et al. (2016) or Merz and Blöschl (2009) and is recommended if a clear distinction between events and low-flow is important. If identification of events is either not possible or relevant, both the obs and sim time series can be treated as two single, long, matching events, and the steps of segment identification and



matching as described in the next section are applied to the entire time series. Despite its simplicity,
this approach has been shown to work well.

### 2.3 Identification, matching and coarse-graining of segments

This section describes the core of the $SD$ concept, i.e. the way to identify, within a matching pair
of an observed and a simulated event, hydrologically comparable points of the hydrographs in order
to quantify their distance in magnitude and time. This procedure has been substantially improved in
the new version of $SD$ and is therefore described here in full.

The term "hydrologically comparable" relates to how a hydrologist would visually compare hydrographs and includes several aspects and constraints: The first constraint is based on the perception
that even if hydrological simulations may deviate from the observations in magnitude or timing, their
temporal order is usually correct. Therefore in $SD$ matching points are compared by preserving their
temporal occurrence: The first point in obs is compared to the first in sim, the second to the second,
the last to the last. Please note that this does not require the two events to be of equal length, as
in $SD$, the hydrograph is considered a polygon from which the points to compare can be sampled
by linear interpolation without restriction to its edge nodes. This is explained in detail below. The
second constraint relates to the slope of the hydrograph: To ensure hydrological consistency, points
within rising segments of sim are only compared to points in rising segments of obs (the same applies to falling segments). This creates a problem related to the within-event variability of the two
hydrographs: It is easy to imagine a case where the number of segments in the obs and sim event differs. This can be either due to sensor-related high-frequency micro fluctuations of the observations,
which can create sequences of many short rising and falling segments, or general deviations of the
simulation from the observation, such as a double-peaked simulated event while the observed event
is single-peaked. In visual hydrograph evaluation, a hydrologist will detect the dominant patterns of
rise and fall in the two time series and identify matching segments by doing two things: Filtering out
short, non-relevant fluctuations and then relating the remaining by jointly evaluating their similarity
in timing, duration and slope. The stronger the overall disagreement of the obs and sim event, the
more visual "coarse-graining" will be done before the hydrographs are finally compared, while at the
same time the degree of coarse-graining will also influence the hydrologist's evaluation of the hydrograph agreement: The higher the required degree of coarse-graining, the smaller the agreement.
In $SD$, these steps are emulated by iteratively maximizing an objective function: While increasingly
coarse-graining the two events, their overall time and magnitude distance is evaluated. The final evaluation of agreement is then done on the level where the optimal trade-off between coarse-graining
and hydrograph distance occurs, i.e. where the objective function is minimal. The procedure consists
of four steps and is explained in the following sections: (1) determination of segment properties, (2)
equalizing the number of segments in the obs and sim event, (3) iterative coarse-graining and (4)
distance computation for the optimal coarse-graining level.




1. For each segment $i$ in the initial sequence of rises and falls of an event, its properties relevant for coarse-graining are determined: Start and end time, duration ($dt(i)$) and absolute magnitude change ($dQ(i)$). From this the relative duration ($dt^*(i)$), and the relative magnitude change ($dQ^*(i)$) of each segment is calculated, i.e. its duration normalized by the total duration and its magnitude change normalized by the total sum of absolute magnitude changes

of the entire event. $dt^*(i)$ and $dQ^*(i)$ are then used to determine the relative importance of each segment ($I_{SEG}(i)$) using the euclidean distance Eq. (1). Taken together, all $I_{SEG}(i)$ of the time series sum up to 1, and segments that are relevant, i.e. that are either very long and/or include large discharge changes receive large values of $I_{SEG}$.

$$I_{SEG}(i) = \sqrt{dt^{*\,2}(i) + dQ^{*\,2}(i)} \qquad (1)$$

2. If the number of segments in the obs and sim event differs, they are (logically) equalized by removing the required number from the event with the surplus. This is done with a directed, iterative aggregation of segments: The least relevant segment (the one with the smallest value of $I_{SEG}$) is selected and assimilated by its two neighbouring segments. For instance, a small relevant rising segment will then be combined with its preceding and succeeding falling segment

to a single, long, falling segment. For the new segment the properties are then determined; its relative importance is the sum of the previous three segments.

It is important to note that this procedure is a purely logical assimilation: Timing and magnitude of the points in the dissolved segment remain unchanged, they are only reassigned to the new and larger segment. This also implies that the meaning of "coarse-graining" in the context

of $SD$ is slightly different from its meanings in Statistics and Thermodynamics or in upscaling (Attinger, 2003; Neuweiler and King, 2002). In the first case, coarse-graining is synonymous with aggregation and averaging of physical quantities, in the second it is related to the preservation of heterogeneity effects upon aggregation. In the case of $SD$, it means that logical (ordering) properties are aggregated, while the absolute values of timing and magnitude of the

data are not changed.

Obviously, this procedure includes a false classification: The rising segment in the previous example is now hidden within a larger falling segment (or vice versa, if a falling segment is dissolved). This can be considered as the "price of coarse-graining" and can be quantified by the number of falsely classified edge nodes ($n^*_{mod}$) of the time series. Therefore $n^*_{mod}$ is a

useful quantity to punish excessive coarse-graining in the objective function, Eq. (2).

3. With the number of segments in the obs and sim events equalized, their $SD$ timing and magnitude distance can be computed. To this end, the first obs segment is compared to the first sim segment, the second to the second, etc. Since the segments can differ in length we here assume that for each segment pair, the appropriate number of points is evenly distributed along



the segment duration and can thus be found by linear interpolation between the time series
      edge nodes. The first point in the obs segment is then connected to the first point in the sim
      segment, the second to the second etc. For each connector its horizontal and vertical projection
      (length in time and magnitude, respectively) is determined (compare again Fig. 1), yielding
      the joint time and magnitude error of the particular point pair.

In the initial version of $SD$, the number of points for each segment pair was found by calcu-
      lating the mean of the two relative durations, $I_{dt}^*$, such that long segment pairs received many
      points and the overall number of connector points of the time series equalled its number of
      edge nodes. In order to better emulate a hydrologist's perception of segment importance, in
      the current version of $SD$ the number of points is determined by the mean relative importance
$I_{SEG}$ (Eq. (1)), of a segment pair. This assigns more points to (and hence puts more emphasis
      on) short but steeply rising segments while still preserving the same overall number of points.

      At this point the result of the $SD$ procedure - a 2-dimensional distribution of time and mag-
      nitude errors, separately for the rising and the falling segments - is available. However, in
      practice often the problem of non-intuitive segment matching spoils the results. Due to the
constraint of time-ordered segment matching, any minor change in monotony within a rising
      or a falling limb that is only present in either the obs or sim event will produce a false matching
      of segments. The left panel in Figure 2 illustrates this problem, where the first falling segment
      in the observed series (labelled by "2" in a square) corrupts segment matching: In chrono-
      logical terms the steep flood rise in obs ("3" in a square) would be compared to the second
rising segment in sim ("3" in a circle), which is obviously wrong. In this case, the $SD$ time and
      magnitude distances will be very large, while visual comparison would most likely be done as
      shown in the right panel of Fig. 2 and yield good agreement.

– Figure 2: Sketch logical aggregation process –

      We overcome this problem using iterative coarse-graining again: Within the events, succes-
sively more segments are (logically) aggregated with their neighbours until finally the entire
      event consists of only two segments: one rise and one fall. Compared to the last step where
      we apply coarse-graining to either sim or obs in order to equalize the number of segments
      in the simulated and observed event, we here apply it simultaneously to the obs and sim
      event. Hence, an equal number of segments and unique segment matching is assured. The
final comparison of the two events is done for the coarse graining step where the total $SD$
      errors and the degree of coarse-graining together are small. Both requirements are considered
      in the coarse-graining objective function ($\theta$). The latter consists of four criteria: i) The num-
      ber of edge nodes in falsely classified segments ($n_{mod}^*$), ii) the cumulated importance of the
      dissolved segments ($I_{SEG,cum}^*$). As discussed above, the false classifications inevitably occur
during the aggregation of segments. Both criteria monotonically increase with the number of




dissolved segments and therefore punish excessive coarse-graining. Further criteria are iii) the $SD$ timing ($E_{SD,t}^*$) and iv) magnitude errors ($E_{SD,Q}^*$) summed up over all segments of the event. They are small when segments that are hydrologically similar, i.e. close in time, duration and magnitude, are compared. As in Eq. (1), each criterion is first normalized to the range of [0 1] and then combined using the euclidean distance Eq. (2):

$$\theta = \sqrt{\gamma_1\, n_{mod}^{*\,2} + \gamma_2\, I_{SEG,cum}^{*\,2} + \gamma_3\, E_{SD,t}^{*\,2} + \gamma_4\, E_{SD,Q}^{*\,2}} \qquad (2)$$

$\theta$ also includes weighting factors ($\gamma_1 \ldots \gamma_4$) for each criterion, which allows user-specific adjustment of the objective function. For example if the temporal agreement of segments is important, the weight for $E_{SD,t}^*$ should be large. As can be seen in Fig. 2, dissolving a single segment can drastically change the events' overall $SD$ time and magnitude distance. Also, as during the successive removal of segments in coarse-graining, it is impossible to predict which combination of segments dissolved in obs an sim will yield the best value of $\theta$, thus all possible combinations are tested and the best is kept. If e.g. both the obs and sim event consist of 10 segments, 10 x 10 combinations of segment dissolutions are tested (obs 1 with sim 1, obs 1 with sim 2, etc.). The coarse-graining scheme is thus computationally demanding. The combination with the minimum $\theta$ is kept and serves as the basis for the next segment reduction step in the coarse-graining procedure.

4. Once the coarse-graining is done, the optimal value of $\theta$ is available for each reduction step, starting with the initial number of segments and ending with two. In Fig. 3, this is shown for a 3-peak event with initially 15 segments. As can be seen in the lower right panel, the value of the objective function is initially high: Here segment matching is poor and as a result $SD$ timing errors and thus $\theta$ are high (upper left panel). After dissolving three segments, agreement is much better (lower left panel) and $\theta$ is at its minimum. Further segment aggregation does not further decrease $SD$ errors, but now the number of falsely classified nodes increases and leads to an increase of $\theta$ (upper right panel). The interplay of the two antagonist parts of $\theta$ often leads to the occurrence of a local minimum. The related reduction step can then regarded as the optimal degree of coarse-graining and the final values of $SD$ time and magnitude errors are determined based on this level.

– Figure 3: Coarse-graining –

## 2.4 Modifications in the $SD$ error model

In the initial version of $SD$, the magnitude error ($E_{SD,Q}$) was calculated as the absolute difference between points in sim and obs linked by a Series Distance connector (c):

$$E_{SD,Q}(c) = Q_{obs}(c) - Q_{sim}(c) \qquad (3)$$



In the current version, the magnitude error can alternatively be scaled by the mean of the connected

points:

$$E_{SD,Q}^*(c) = \frac{Q_{obs}(c) - Q_{sim}(c)}{\frac{1}{2}(Q_{obs}(c) + Q_{sim}(c))} \tag{4}$$

This yields a scaled (dimensionless) expression of the vertical error ($E_{SD,Q}^*$) which facilitates the

construction of uncertainty ranges of variable width (see chapter 3). As in the first version of $SD$,

both absolute and relative vertical error values $E_{SD,Q}^{(*)} > 0$ indicate that $Q_{obs}(c) > Q_{sim}(c)$. The

calculation of series distance timing errors ($E_{SD,t}$) according to Eq. (5) remained unchanged. Error

values of $E_{SD,t} > 0$ indicate that obs occurs later than sim:

$$E_{SD,t}(c) = t_{obs}(c) - t_{sim}(c) \tag{5}$$

Similar to the scaling of the vertical error, the timing error could also be scaled using e.g. event

duration. This could be helpful if the error compared to the length of the event (or the average length

of all events in the time series) is of interest.

The application of $SD$ timing and magnitude error models ($E_{SD,t}(c) \; and \; E_{SD,Q}(c)$) makes

sense where timing errors are both present and detectable, i.e. during events where discharge is

not constant in time. During low-flow conditions time offsets are however difficult, if not impossible

to detect. Therefore, a simple one-dimensional, vertical, "standard" error model analogous to Eq.

(3), which relates values at the same time step $t$ suffices here:

$$E_S(t) = Q_{obs}(t) - Q_{sim}(t) \tag{6}$$

Analogously to the scaled vertical $SD$ error model in Eq. (4), a scaled version of the one-

dimensional vertical error model ($E_S^*(t)$) was added:

$$E_S^*(t) = \frac{Q_{obs}(t) - Q_{sim}(t)}{\frac{1}{2}(Q_{obs}(t) + Q_{sim}(t))} \tag{7}$$

## 3 Error dressing: A heuristic approach for the construction of uncertainty ranges

The $SD$ concept can be applied to a variety of tasks such as model diagnostics, parameter estima-

tion (calibration) or the construction of uncertainty ranges. In this section we provide one example

thereof and describe a heuristic approach for the construction of uncertainty ranges for deterministic

streamflow simulations. Uncertainty ranges provide regions of confidence around an uncertain es-

timate, are of practical relevance and a straightforward means to highlight and to assess magnitude

(and timing) uncertainties of hydrological simulations or forecasts. Conceptually, uncertainty ranges

should be wide enough to capture a significant portion of the simulated values but as narrow as pos-

sible to be precise and, thus, meaningful. These requirements are antagonistic as large uncertainty

ranges, which capture most or all observations, are usually imprecise to a degree that makes them

useless for decision-making purposes (Franz and Hogue, 2011).





The method we propose here follows the concept proposed by Roulston and Smith (2003) and yields quantitative estimates of forecast uncertainty by "dressing" single forecasts with historical error statistics. The original approach was designed to dress ensemble forecasts; for $SD$ it was adapted to deterministic stream flow simulations and extended from one dimension (magnitude) to two (magnitude and timing). Like statistical approaches to uncertainty assessment, error dressing is based on the fundamental assumptions of ergodicity and stationarity, i.e. the assumption that errors that occurred in the past are reliable predictors for errors in the future. In the following we first outline the regular, one-dimensional deterministic error dressing method and then describe its modifications for $SD$.

### 3.1 The one-dimensional case

Provided with a record of past streamflow observations ($O_{hist}$) and corresponding model simulations ($S_{hist}$), any valid error model such as Eq. (6) can be applied to calculate a distribution of historic errors. This distribution can then be sampled (Fig. 4, upper left panel) using a suitable strategy and the selected subset of errors can be applied to each time step of the simulation. Connecting all upper and all lower values of the dressed errors yields corresponding envelope curves (Fig. 4, upper right panel). For this procedure Roulston and Smith (2003) coined the term "error dressing".

Figure 4: heuristic concepts on sampling strategy and construction of uncertainty envelopes for both, the one- (upper row) and two-dimensional case (lower row).

The choice of the sampling strategy, however, strongly influences the statistics of the resulting uncertainty ranges and should be carefully selected. In our case, the precondition was that the approach should be extendible to two-dimensional cases to allow its later application to the error distributions of the $SD$ approach. Therefore we defined the sampling strategy according to the "variance contribution" which is straightforward to apply for the one-dimensional case: For each point of the error distribution its relative contribution ($d\sigma_i^2$) to the (unbiased) variance of the total error distribution ($\sigma_x^2$) is calculated according to Eq. (8):

$$d\sigma_i^2 = \frac{(x_i - \bar{x})^2}{n\,\sigma_x^2}\,100 \tag{8}$$

Here $\bar{x}$ and $n$ denote the mean and the size of the corresponding error distribution. The usage of the unbiased variance (having $n$ in the denominator not $n-1$) ensures that all $d\sigma_i^2$ sum up to 1. Next, all points of the error distribution are ordered by the values of $d\sigma_i^2$, and, starting with the smallest, a desired subset of all $d\sigma_i^2$ (e.g. 80 %) is taken from the list. This subset represents an (informal) probability ($p \in [0\ 1]$) as it relates to the number of observations that fall within the uncertainty range. Small values of $p$ are associated with narrow (sharp) uncertainty ranges, but at the cost of a higher portion of true values that fall outside. Contrary, high values of $p$ cause wide (imprecise) uncertainty ranges which however contain most errors that occurred in the past. For practical applications, typi-





cally coverages of 80 to 90 % are chosen. In Fig. 4, top left panel, the coverage (range between the upper and lower coverage lines) was set to $p = 0.8$.

### 3.2 The two-dimensional case

$SD$ yields 2-dimensional distributions of coupled errors in timing and magnitude and thus requires a 2-dimensional strategy for the sampling of error subsets and the construction of envelope curves

(Fig. 4, lower row).

How to sample from bivariate distributions of coupled errors with different units? Statistics and computational geometry offer concepts based on ordering of multivariate data sets, such as "geometric median" or "centerpoint" approaches. The former provides a central tendency for higher dimensions and is a generalization of the median which, for one-dimensional data, has the property

of minimizing the sum of distances. Centerpoints are generalisations of the median in higher dimensional Euclidean space and can be approximated by techniques such as the "Tukey depth" (Tukey, 1975) or other methods of depth statistics (Mosler, 2013). Here, however, we want the errors to be centred around the mean (not around the median). Hence we apply the same concept that we use for the one-dimensional case to $SD$ in that we sample based on the combined contribution of each point

to the total variance. Analogously to Eq. (8) we calculate the relative timing ($d\sigma_t^2$) and magnitude ($d\sigma_Q^2$) contribution of each point to the total variances of the corresponding distributions. Their sum yields an estimate of the combined contribution of each point to the combined variance of both error distributions:

$$d\sigma_{t+Q}^2 = d\sigma_t^2 + d\sigma_Q^2 \tag{9}$$

Analogously to the one-dimensional case, the points are ordered by increasing combined variance contribution $d\sigma_{t+Q}^2$ and, starting from the point with the smallest value (which is close to or at the mean), a subset of errors can be extracted. The shape of the resulting subset depends on the underlying distribution of errors. Uncorrelated errors yield more or less circular/ oval shapes (Fig. 4, lower left panel). Contrarily, correlated errors yield different shapes which is valuable for diagnostic

purposes.

$SD$ distinguishes periods of low-flow, rising and falling limbs. Hence subsets of two 2-d error distributions (rising and falling limb) and from one 1-dimensional error distribution (low-flow) are calculated and applied to each point of a simulation: Points of low-flow are dressed with the low-flow error subset, points of rise with error subsets from rising limbs etc. Altogether this yields a region of

overlapping error "ovals" around a simulation (Fig. 4, lower right panel), which can for convenience be represented by an upper and lower envelope curve. These lines are found by subdividing the time series into time slices of length $dt$ (the temporal resolution of the original series), centred around each edge node of series. In each time slice, the magnitude and timing of the largest and smallest error are identified. These values span the upper and lower limit of the uncertainty envelope, respectively.





Using linear interpolation yields the upper and lower limits of the envelope at the points in time of
the original series, which is useful to calculate evaluation statistics.

## 4 Case study

This case study, based on real-world data, serves to present and to discuss relevant aspects of $SD$ by
comparison with a benchmark error model ($BM$).

### 4.1 Data and site properties

We used discharge observations ($O_{hist}$) of a 6-year period (30.10.1999-30.10.2005) from gauge
"Hoher Steg" (HOST), which is located in the small alpine catchment of the Dornbirner Ach river
in North-Western Austria. Catchment size is 113 $km^2$, the elevation range is 400-2000 m.a.s.l. and
mean annual rainfall differs between 1100 and 2100 $mm\ yr^{-1}$. For the 6-year period period, hourly
hydro-meteorological time series ($n = 52633$ time steps) were used to drive an existing, calibrated
conceptual water budget model of type "LARSIM" (gridded version, resolution = 1 $km^2$, (Ludwig
and Bremicker, 2006)), which yielded acceptable simulations ($S_{hist}$) with a NASH of 0.78. Please
note that for the discussion of the $SD$ concept, neither the model itself nor the catchment properties
are particularly relevant. The main purpose of the case study was to apply realistic data. This is also
the reason why we used the entire 6-year period to both derive and apply the error distributions, i.e.
we did not distinguish periods of error analysis and error application.

### 4.2 Conceptual setup

For the benchmark model, we derived distributions of 1-d vertical errors. We did not differentiate
cases of low-flow and events, which is rather simplistic but standard practice. For the $SD$ approach
we did differentiate these cases. This may be considered an unfair advantage for $SD$ as it allows the
construction of more custom-tailored uncertainty envelopes. However, as the objective of the case
study is not a competition between the two approaches but a way to present interesting aspects of
$SD$, we considered it justified. For $SD$, the required starting and end points of hydrological events
were manually determined both in $O_{hist}$ and $S_{hist}$ by visual inspection. Altogether there were n=123
events in each series and they were fully matching, i.e. no missing events or false alarms occurred.
As obviously the resulting contingency table is trivial, it is not further discussed here.

Both for $SD$ and $BM$ we applied scaled errors ($E^*_{SD,Q}(c)$ according to Eq. (4) and $E_{BM}$ accord-
ing to Eq. (7), respectively), as we found that compared to the standard error model, they are more
applicable across the usually large discharge ranges present in hydrographs. For $SD$, the weights
$\gamma_1, \ldots, \gamma_4$ used in the objective function of the coarse-graining procedure (Eq. (2)) were set to 1, 1,
5 and 0, respectively, based on iteratively maximizing the visual agreement of segments in matching
events of sim and obs. Additional studies with different data sets (not shown here) yielded simi-





lar optimal weights, which corroborates that this is a relatively robust choice and sufficient for a
proof-of-concept, as intended in this study. For more widespread applications, a detailed sensitivity
analysis is desirable.

For both approaches we derived empirical error distributions from the entire test period and then
used them, in the same period, to construct uncertainty envelopes around the simulation $S_{hist}$ using
the error dressing approach as described in chapter 3. To ensure comparability we enforced identical
coverages for both approaches during the construction of the envelope curves, i.e. we made sure that
the desired fraction of observations (e.g. 80 %) fell within the uncertainty envelope. For the standard
error model this was straightforward: If from the 1-d distribution of errors a subset of $p = 80$ % is
selected and used to construct the uncertainty envelope as described in 3.1 for the same period of
time, then by definition the number of observations within the envelope must also be 80 %. For $SD$
however, as a consequence of error ovals overlapping in time (Fig. 4, lower right panel), this is not
self-evident and typically many more observations fall within the uncertainty envelope than the level
$p$ at which the subset of the 2-d error distribution is sampled. This issue was solved by iteratively
sampling the error distributions at various levels of $p$ until the desired percentage of observations
(here: 80%) fell within the uncertain envelope.

### 4.3 Evaluation of deterministic uncertainty ranges

The evaluation of deterministic uncertainty ranges requires methods to quantify properties such as
coverage or precision. Here we propose a set of statistics which can be applied to uncertainty ranges
irrespective of how they were constructed. While this ensures comparability of the $SD$ and $BM$
derived ranges, it does not exploit the advantages of the $SD$ approach, i.e. separate treatment of time
and magnitude uncertainties.

1. Coverage ($\phi$) is the most intuitive criterion. It quantifies the ratio of observations that fall
   inside the simulated uncertainty range and can take values between 0 (not a single observed
   value included) and 1 (all observations included). $\phi$ can easily be obtained as the number of
   observations ($n_{obs}$) that fall inside the uncertainty range around a simulation, divided by the
   total length of the time series ($n$):

$$\phi = \frac{n_{obs}}{n} \tag{10}$$

2. Precision ($PRC$) allows comparison of different uncertainty ranges. $PRC$ is the average
   width of the uncertainty envelope, i.e. the average difference of the upper ($UE^+(t)$) and the
   lower ($UE^-(t)$) envelope curve. The smaller $PRC$, the sharper the uncertainty range. High
   coverages $\phi$ typically require wide uncertainty ranges and thus, high values of $PRC$. $PRC$
   has the same unit as the discharge time series.

$$PRC = \frac{1}{n} \left( UE^+(t) - UE^-(t) \right) \tag{11}$$





3. Finally we suggest scaling $PRC$ by the value of the simulation according to Eq. (4), i.e. to express uncertainty relative to the magnitude of the simulation. $PRC^*$ is dimensionless and decreases with decreasing width of the uncertainty range. An uncertainty range of zero width yields a $PRC^*$ of zero. Hence, small values of $PRC^*$ indicate high skill.

$$PRC^* = \frac{1}{n} \frac{(UE^+(t) - UE^-(t)}{Q_{sim}(t)} \tag{12}$$

In the case study, we used $\phi$ as a means to ensure comparability rather than for comparison: Coverage for both the $SD$ and $BM$ approach was set to $80 \pm 0.5$ %. For $SD$ the required percentage of sampled errors was found by trial and error to be $p = 76$ % (Table 2). With coverage equalized, $SD$ and $BM$ can be directly compared by $PRC$ and $PRC^*$. High (relative) precision (i.e. small values of $PRC^{(*)}$) indicate better performance. If the evaluation of uncertainty ranges with respect to over- and undershooting is of interest, additionally the percentage of observations above or below the uncertainty range can be computed analogously to Eq. (10). This is for instance of interest for flood forecasters (who try to minimize overshooting) or water supply managers (who try to minimize undershooting). For the sake of brevity, this has not been further considered here.

## 5   Results and discussion

In this section we first discuss some general aspects of the $SD$ concept and then compare it to the benchmark approach using the case study data.

### 5.1   Potential and limitations of the core $SD$ concept

Series Distance is an elaborate method for the comparison of simulated and observed streamflow time series. The concept allows the distinction between different hydrological conditions (low-flow, rising and falling limbs) and determines joint errors in timing and magnitude of matching points within matching segments of related hydrographs. Differences in the high- and/or low-frequency agreement of the obs and sim hydrographs are considered with an iterative "coarse-graining" procedure, which effectively mimics visual hydrograph comparison. This differentiated evaluation makes $SD$ a powerful tool for model diagnostics and performance evaluation.

The challenges of $SD$ are however in the details: The robust, precise and meaningful partitioning of the hydrograph into periods of low-flow and events is difficult. We tested various approaches including baseflow separation and filtering techniques (e.g., Douglas and Peucker, 1973; Chapman, 1999; Perng et al., 2000; Eckhardt, 2005), penalty functions (Drabek, 2010), fuzzy logic (Seibert and Ehret, 2012), and the methods proposed by Merz and Blöschl (2009) and Norbiato et al. (2009). In all cases, the results were unsatisfactory when applied to a range of different flow regimes. The same applies for the matching of conjugate events in obs and sim. Currently, there is no robust and automated method available for any of the two cases. Possible remedies are the adaptation of any





of the methods proposed above to specific conditions (Seibert et al., 2016), manual event detection
and matching or to treat the entire time series as a single, long event, at the expense of losing the
separate treatment of low-flow cases. Within an event, the quality of the segment matching largely
determines the quality of the subsequent matching of obs and sim points and hence the quality of
the $SD$ error calculation. This challenge has been solved in a mostly very satisfactory way by the

iterative coarse-graining procedure. The resulting set of matching segments and the required degree
of coarse-graining is in itself a useful result which can be used for comparative hydrograph analysis.
The objective function can be tailored to different requirements by modifying $\gamma_1 \ldots \gamma_4$ in Eq. (2).
We found that the configuration presented in the case study (chapter 4.2) which emphasizes timing
errors ($E^*_{SD,t}$) generally produces good agreement with visual coarse-graining and we thus suggest

it as default parametrization. A more in-depth sensitivity analysis of $\gamma_1 \ldots \gamma_4$ using streamflow data
from different regimes would however be desirable.

The hydrograph matching algorithm ($HMA$) proposed by (Ewen, 2011) is, to our knowledge,
the only method which is similar to the $SD$ concept in the sense that it relates elements of an ob-
served to elements in a simulated hydrograph in an intuitive manner. Similar to $SD$, the $HMA$ uses

connectors ("rays") to establish these relationships. However, the manner in which these connec-
tors are identified is different. The $HMA$ moves chronologically through all elements of obs and
calculates the distance to points in sim which are located within a defined window around the ele-
ment in obs using a penalty function. This procedure generates a (possibly huge) matrix of penalty
values. In a second step the optimal "path" through this matrix is identified which yields the con-

nectors. This makes the $HMA$ computationally demanding. However, the same also applies for $SD$
as the coarse graining scheme may require a large number of iterations. The advantage of $SD$ is
that unique relationships of points in obs and sim are established, which is not the case for $HMA$.
Leaving aside these methodological finesses, we believe that for hydrological studies there is a large
potential for "intuitive" distance metrics which is not yet fully exploited: In the inter-comparison

study of Crochemore et al. (2014) both $HMA$ and $SD$ closely resembled expert judgement and out-
performed standard (vertical) distance metrics during high and, for $HMA$, also low-flow conditions.

### 5.2  Potential and limitations of the error dressing method

Error dressing is a simple method and straightforward to apply. Conceptually it is very similar to
statistical concepts like the total uncertainty method introduced by Montanari and Grossi (2008) in-

sofar as it does not distinguish between different sources of uncertainty. Unlike rigorous statistical
concepts, error dressing however does not make any assumptions on the nature of the population
of errors: They are directly sampled from the empirical distribution, thus avoiding the need to fit
a theoretical distribution to the data. The fundamental assumption of error dressing is hence that
the available sample represents the population and implies that the skill of the resulting uncertainty

ranges strongly depends on the representativeness of the empirical distribution of errors. This may



not be the case if records are short and/or if the available data only cover a limited range of conditions. This is however a frequent problem of statistical methods for uncertainty assessment (not only in Hydrology), where often the extremes are of interest, although they are rare by definition (Montanari and Grossi, 2008). Further uncertainties arise from erroneous observations, which is a

common problem in Hydrology. These conceptual limitations lead to the fundamental question of whether it is better to profit from statistical (or heuristic) information on the basis of the stationarity assumption, or to neglect it by questioning the assumption itself (Montanari, 2007). This discussion is however beyond the scope of this study.

The error dressing concept in the presented form does not distinguish between seasonality or
different flow magnitudes as the same error distributions are applied to each rising (and/or falling) limb. More sophisticated implementations are of course possible, such as a differentiation of errors according to flow magnitudes to better capture extremes, or differentiation according to forecast lead times. The same applies for the sampling strategy: As an alternative to the method presented here based on combined variance contribution, sampling of specific quantiles using the median as
central reference or fitting and application of any parametric function to the distribution is of course possible. A practical insight from applying the error dressing concept is that the variance-based method effectively filters outliers, which sometimes occur when errors are calculated between poorly matching segments.

A last general issue relates to the sampling from the two-dimensional error distribution. Due to
the superposition of error clouds in successive time steps it is possible that errors in timing at one time step "mimic" errors in magnitude at neighbouring time steps (Fig. 4, bottom right panel). This depends on the temporal extent of the error "ovals". As a consequence, the relationship between $p$, which defines the size of the subset from the distribution, and coverage ($\phi$) becomes non-unique. In any case it is not directly linear as in the one-dimensional case where $p$ equals $\phi$ per definition (at
least for the period of calibration). Typically $\phi$ exceeds $p$ in the two-dimensional case, and desired coverage rates of $\approx 80\%$ require us to set $p$ to $\approx 0.65 - 0.75$. If a specific coverage is desired, the related value of $p$ is best found by iteration. Altogether, the error dressing concept seems suitable for practical applications where long time series are available but more sophisticated uncertainty assessments are not feasible, either because of the required effort or because of limited knowledge
of the underlying system.

### 5.3   Case study results

As described in chapter 4.2, within the 6-year time series altogether n=123 events were manually identified in both obs and sim. The events matched perfectly, i.e. no missed events or false alarms occurred. This is often the case for simulations of responsive catchments where rainfall events trigger
runoff events in most cases and where the precipitation time series thus carries important information about the occurrence of hydrological events. This is not necessarily the case for hydrological



forecasts, especially mid- to long-term, where false precipitation events can generate false hydrological events. In the latter case, event-based information contained in the contingency table can be valuable. The mean event durations were 146 and 154 hours for obs and sim, respectively, and on average each event initially contained 13 (sub)peaks. The optimal level of event comparison was on average achieved after two coarse-graining steps, which reduced the number of peaks on average to four and led to average durations of 37 hours for rising limbs and 109 hours for falling limbs for both obs and sim. These statistics again bear diagnostic potential as they can be interpreted as surrogates for the mean concentration time of the catchment or as a reservoir constant and can thus be compared to other data. Generally, the matching of segments resulting from the coarse-graining procedure corresponded well with visual human reasoning (not shown). In the following we compare the error distributions and uncertainty envelopes derived from the $SD$ and $BM$ approach for our test case.

### 5.3.1 Comparison of error distributions

Altogether four error distributions were calculated: For $SD$ two 2-d distributions (one for the rising and one for the falling event limbs), and one 1-d distribution for the low-flow conditions; for $BM$ a single 1-d distribution of magnitude errors for the entire time series. The distributions are shown in Fig. 5, corresponding statistics in Table 1.

Comparing the 2-d distributions reveals distinct differences in shape: For the rising limbs it is rather oval, for the falling limbs it is almost circular. This is particularly evident in the sampled subsets. The uniform spread of the errors within the oval and the circle indicates that for the data at hand, the timing and magnitude errors are largely uncorrelated, but dependent upon the hydrological conditions (rise or fall). The (scaled) magnitude errors for both distributions are located between $\pm 1.5$. The magnitude biases for both distributions are relatively small and lie, according to the ranges provided by Di Baldassarre and Montanari (2009) within the error of measurement: $SD_{Q,rise} = 0.1$ for the rising limbs, $SD_{Q,fall} = 0.008$ for the falling limbs. Note that positive magnitude biases indicate simulations that on average underestimate the observations. For timing errors, the differences are more pronounced: While for the rising limbs, timing errors are located between $\pm 10$ hours for the sampled subset and biased by -0.2 hours (indicating simulations lagging behind the observations), for the falling limbs both the bias (-3 hours) and the range ($\pm 20$ hours) are much larger. Please note that we discuss the timing errors of the subset here rather than those of the entire sample, as the latter includes few but large outliers caused by occasional poor matching of falling limbs during coarse-graining.

– FIGURE 5: Case study error distributions

The $SD$ 1-d distribution for low-flow, is quite different from the events, significantly biased ($SD_{LF} = -0.35$) and the 80 % subset ranges from -0.89 to 0.19. This means that low-flow simulations gener-



ally overestimate the observations by 35 % on average, and that overestimations can be more extreme
than underestimations. For $BM$, the distribution shows a negative bias ($BM$=-0.23) and the 80 %
subset of errors ranges from -0.83 to 0.37. The lower tail is thus comparable to the $SD$ distribution,
although the latter is based on low flow only and the $BM$ distribution is based on the entire time se-
ries, including the events. The difference in the upper tail is however almost 20 % indicating higher
errors in the $BM$ case than in the $SD$ case. The apparent similarity in the lower tail is due to the
dominance of low flow conditions in the time series: From altogether 52633 hourly time steps, fully
two-thirds belong to low flow conditions, which means they dominate the distribution.

– Table 1: Statistics of the error distributions

Together, these results confirm that different flow conditions (low-flow, rising or falling limbs of
events) exhibit different error characteristics. This suggests that a differentiation of "hydrological
conditions" can be meaningful. For instance, timing errors of the recession in the case study would
be strongly underestimated by timing errors of the rising limbs, and vice versa, as depicted in the
lower panel of Fig. 7. The comparison of 1-d distributions of the $SD$ and $BM$ model revealed that
important error characteristics of rare events can be shadowed by frequent but often less relevant low
flow conditions.

### 5.3.2 Comparison of uncertainty envelopes

Subsets of both the $SD$ and $BM$ error distributions were used to construct uncertainty envelopes
($UE$) around the entire simulated time series $S_{hist}$. For better visibility of the details, only a three-
week period is shown in Fig. 6; the envelope statistics presented in Table 2 however are based on
the entire series. The percentages $p = 76$ % for $SD$ and $p = 80$ % for $MD$ of sampled errors in the
subsets were selected such that the overall coverage ($\phi$) of the uncertainty envelopes was 80 % in
both cases. Compared to $UE_{BM}$, the $UE_{SD}$ in Fig. 6 appears both smoother and more "inflated".
This is due to the timing component of the error model which spreads the uncertainty envelope in
time. This is particularly visible at the beginning of the events. Here, timing errors dressed to a given
time step clearly extend to neighbouring time steps, representing the uncertainty about the true event
start. In the case of several peaks occurring within a short time (Fig. 6, last event), the smoothing
effect of the timing component can lead to a merging of the related uncertainty envelopes towards
a single, large region. Also the difference between (smaller) timing errors in the rising limbs and
(larger) timing errors in the falling limbs are visible. Partly, timing errors of the falling limb even
mimic timing errors in the rising limb (compare also Fig. 7, lower panel).

– FIGURE 6: Uncertainty envelopes

In comparison, the uncertainty envelope of the $BM$ model appears slimmer and more precise. How-
ever, due to the lack of consideration of timing uncertainties, especially during steep flood rises, the




uncertainty envelopes become very narrow. Such a "vanishing" of the uncertainty envelopes implies that there are no timing errors to be expected at all (compare e.g. the period 06.-07.06.2001 in Fig. 6), which is deceptive, keeping in mind the $SD$ results for the timing errors (Fig. 5). We thus consider this aspect a disadvantage of the one-dimensional error dressing method, especially as the timing of flood rises are often critical in hydrological applications (Seibert et al., 2014).

The statistical evaluation of the different uncertainty envelopes (Table 2) confirms the visual impression: The $BM$ uncertainty envelope outperforms $SD$ in terms of absolute and relative precision ($PRC$ and $PRC^*$, respectively) given identical coverage ($\phi$). On average, $UE_{SD}$ is 3.1 m$^3$s$^{-1}$ "wider" than the benchmark envelope, which corresponds to a relative difference of 30 % as indicated by $PRC^*$. This suggests that use of the $SD$ concept to construct uncertainty envelopes implies a trade-off of two effects: On the one hand, the explicit consideration of timing errors potentially yields better tailored uncertainty envelopes, as apparent timing errors can be treated as such. On the other hand, if timing is not a dominant or at least substantial component of the overall error, the time-spreading effect of the $SD$ envelope construction can lead to an undesirable inflation effect. In our case study, the latter effect apparently predominated. For hydrological forecasts based on uncertain meteorological forecasts however the opposite may be the case.

– Table 2: Statistics of the uncertainty envelopes

### 5.3.3 Disentangling the importance of magnitude and timing errors

To further investigate the individual effects of errors in timing and magnitude, we also applied them separately to the simulated time series. To this end we used case-specific (low-flow, rising and falling limbs) subsets of 2-d error distributions to each point of the simulated time series just as in the previously described 2-d error dressing approach. The difference was that we did not apply the entire error subset ("oval" or "circle") but its projection on the time and magnitude axis, respectively. The resulting "uncertainty bars" therefore extend from the maximum to the minimum magnitude (upper panel) and timing (lower panel) values of the error subsets and are depicted in Fig. 7. For comparison we also plotted the magnitude errors of the $BM$ approach. In this representation it becomes obvious that the error bars of the $SD$ and $BM$ approach show considerable differences with respect to extent and symmetry. For the magnitude error bars the deviations are most pronounced in the rising limbs and less so in the falling limbs and during low flow conditions. While the $SD$ method reflects the underling characteristics of the errors, the $BM$ method applies the same error to all cases. Constructing an uncertainty envelope from only the $SD$ magnitude errors would yield an envelope comparable to that of $BM$ but be more variable and have higher uncertainty towards overestimations than towards underestimations. Note that the true distribution of errors within the error bars is unknown.

The lower panel in Fig. 7 reveals that the uncertainties with respect to timing are considerable, typically during the recessions. Combining horizontal and vertical errors to construct the 2-d $SD$



uncertainty envelope using the method described in chapter 3 will inevitably cover a large region. While this is undesirable, it points towards possible alternatives to construct uncertainty ranges: Rather than uniting the horizontal and vertical uncertainty components, intersecting them would

also be possible, for example, and most likely narrow the uncertainty envelope. Also, discharge time series usually exhibit considerable autocorrelation, and so do related simulation errors. Exploiting this memory effect by time-conditioned sampling of the error distribution via a Markov process as proposed by would be a further alternative to better tailor uncertainty envelopes (Vrugt et al., 2008; Montanari et al., 1997).

Finally, even if the $SD$ error distributions are not used to construct uncertainty envelopes, knowledge of magnitude and timing error distributions is valuable for model diagnostics: In their approach to identifying characteristic error groups in hydrological time series Reusser et al. (2009) had to inversely infer the effect of timing errors to their signatures; $SD$ offers a method to directly measure timing errors and thus to improve this step.

730         – FIGURE 7: Error bars

## 6   Conclusions

The main goal of this paper was to present major developments in the Series Distance $(SD)$ concept since its first version presented by Ehret and Zehe (2011). These include the development of an iterative optimization procedure which effectively mimics "coarse-graining" of hydrographs when

comparing them visually. The parameters of the inherent objective function were derived manually for this study; for more widespread applications we recommend an in-depth sensitivity analysis. Coarse-graining yields a set of matching segments within observed and simulated hydrological time series and the optimal degree of coarse-graining, both of which can be used as input for comparative hydrograph analysis. Further developments include the introduction of a scaled error model which

has proven to be better applicable across large discharge ranges than its non-scaled counterpart, and "error dressing", a concept to construct uncertainty ranges around deterministic streamflow simulations or forecasts. Error dressing includes an approach to sample empirical error distributions by increasing variance contribution, which we extended from standard 1-dimensional distributions to the 2-dimensional distributions of combined time and magnitude errors of $SD$.

Applying the $SD$ concept and a benchmark model $(BM)$ based on standard magnitude errors to a six-year time series of observations and simulations in a small alpine catchment revealed that different flow conditions (low-flow, rising and falling limbs during events) exhibit distinctly different characteristics of timing and magnitude errors with respect to mean and spread. Separate treatment of timing and magnitude errors and a differentiation of flow conditions (as done in $SD$) is thus

recommended in general as it preserves useful information. Exploiting these characteristics and their correlations can support targeted model diagnostics. Deeper insights can easily be provided if the





error distributions are further differentiated by discharge magnitude classes, season or by considering the temporal autocorrelation of errors. The latter would allow the development of a time-conditioned error sampling strategy when constructing 2-d uncertainty envelopes.

Applying the error distributions of both $SD$ and $BM$ to construct uncertainty ranges around the (fairly accurate) simulation revealed a remarkable timing uncertainty. This suggests that we commonly underestimate the role of horizontal uncertainties in streamflow simulations. For the given data, the $BM$ derived uncertainty ranges were in consequence visually more narrow and statistically superior to the $SD$ ranges. This suggests that use of the $SD$ concept to construct uncertainty

envelopes according to the proposed error dressing method implies a trade-off of two effects: On the one hand, the explicit consideration of timing errors potentially yields better tailored uncertainty envelopes, as apparent timing error are treated as such. On the other hand, the time-spreading effect of the $SD$ envelope construction, which essentially is the union of the time and magnitude error uncertainty ranges, can lead to an undesirable inflation. For the case study data, the latter effect pre-

dominated while for hydrological forecasts based on uncertain meteorological forecasts the opposite may be the case. This also opens interesting avenues for new ways to construct uncertainty ranges based on the $SD$ concept, e.g. as the intersect (rather than the union) of the two error components.

   We conclude that Series Distance is an elaborate concept for the comparison of simulated and observed stream flow time series which can be used both for detailed hydrological analysis and

model diagnostics. Its application however involves considerably more effort than standard diagnostic measures, which is typically justified if timing errors are dominant or of particular interest. More generally, we believe that for hydrological studies there is a large potential for "intuitive" distance metrics such as the hydrograph matching algorithm proposed by (Ewen, 2011) or the $SD$ concept, which should be further exploited as suggested by Crochemore et al. (2014).

To foster the use of the $SD$ concept and the methods therein we publish a ready-to-use Matlab program code alongside to the manuscript under a CreativeCommons license (CC BY-NC-SA 4.0). It is accessible via https://github.com/KIT-HYD/SeriesDistance. This repository also includes extended versions of the $SD$ concept which we did not describe in full length here. These allow for a continuous usage of the method (no data on events required) and/or a differentiation of vertical

errors according to flow magnitude.

*Acknowledgements.* We thank Tilmann Gneiting from the Heidelberg Institute for Theoretical Studies (H-ITS) for valuable discussions on the error dressing concept, Clemens Mathis from Wasserwirtschaft Vorarlberg for providing the case study data and hydrological model and all users of $SD$ who provided valuable feedback and constructive criticism throughout recent years. We also acknowledge support for open access publishing by

the Deutsche Forschungsgemeinschaft (DFG) and the Open Access Publishing Fund of Karlsruhe Institute of Technology (KIT).



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





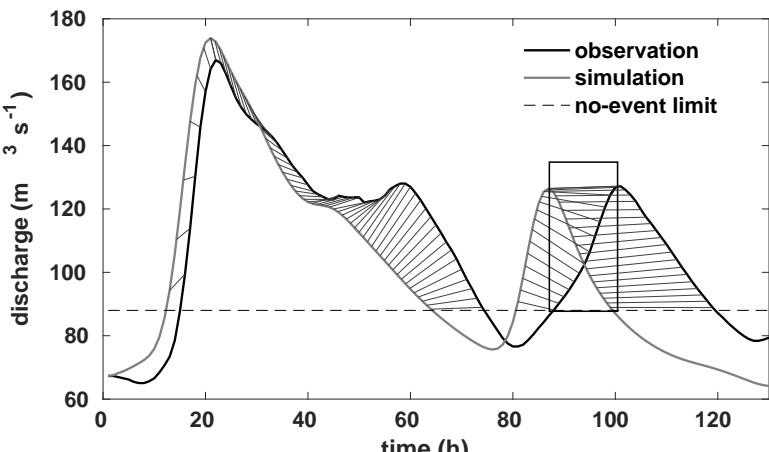

**Figure 1.** Time series of observed (black) and simulated (grey) discharge during a "hydrological event". The horizontal line represents a user specific threshold which differentiates between event and non-event periods. The light grey lines represent the series distance connectors linking hydrologically comparable points in the two time series. Time and magnitude distances are calculated between these points. The black rectangle highlights time steps were a part of the recession of the simulation overlaps with a rising part of the observation (figure from Ehret and Zehe (2011)).

.





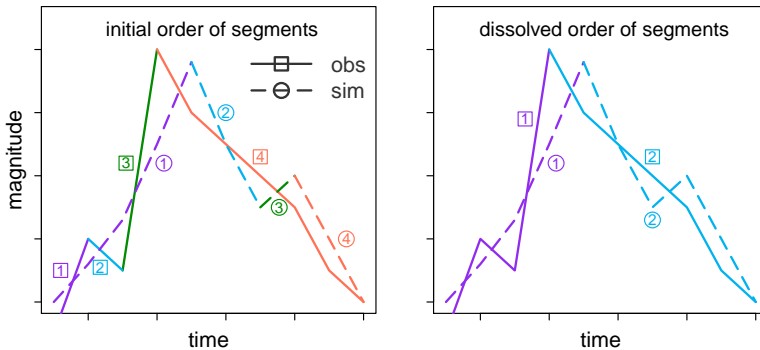

**Figure 2.** Illustration of the time-ordered matching of segments in the coarse-graining procedure. The rising and falling segments of the simulation (sim) and observation (obs) are numbered and colour-coded according to their chronological order. Series Distance compares segments with identical number/colour.



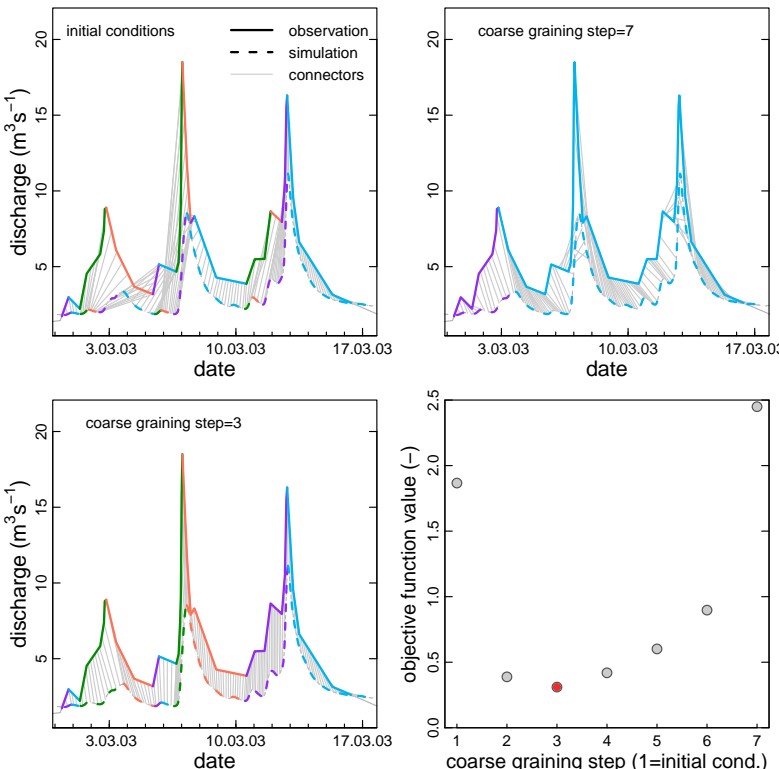

**Figure 3.** Coarse-graining steps: All plots contain data from the same multi-peak discharge event, but for different levels of coarse-graining. The initial conditions (top left) are characterized by a large number of poorly matching simulated (dashed) and observed (solid) segments as indicated by the non-intuitively placed $SD$ connectors (grey lines). Segments required to match according to the chronological order constraint of $SD$ are indicated by matching colours. In the last coarse graining step (top right) the connectors are placed more meaningfully but the representation of the entire event by only two segments (one rise, one fall) appears inadequately coarse. The optimal level of coarse-graining, here reached at step three (bottom left), yields visually acceptable connectors while preserving a detailed segment structure (bottom left). This step is associated with a minimum of the coarse-graining objective function (Eq. (2)), indicated by the red dot in the bottom right panel. Grey dots indicated the values of the objective function for all other coarse-graining steps.





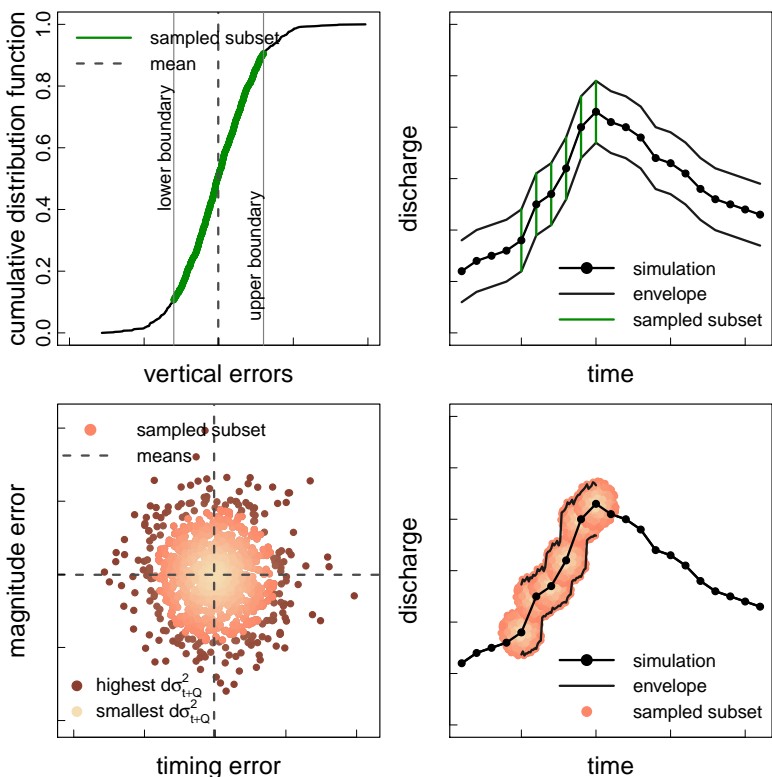

**Figure 4.** Sketch of the one- and two-dimensional error dressing method using normally distributed random numbers (n=1000). The **upper row** shows the one-dimensional case with an empirical cumulative distribution function of errors (upper left panel) and an 80% subset thereof sampled according to increasing variance contribution. The application (dressing) of the subset of errors to a hydrograph and the construction of the corresponding envelop curves is illustrated in the upper right panel. The **lower row** shows the same procedure for the two-dimensional case. From the two-dimensional distribution of empirical errors (bottom left panel) again 80 % (colour coded) are sampled according to the combined variance contribution of both distributions (colour ramp). The bottom right panel contains a sketch of the two-dimensional error dressing method and the construction of envelope curves.





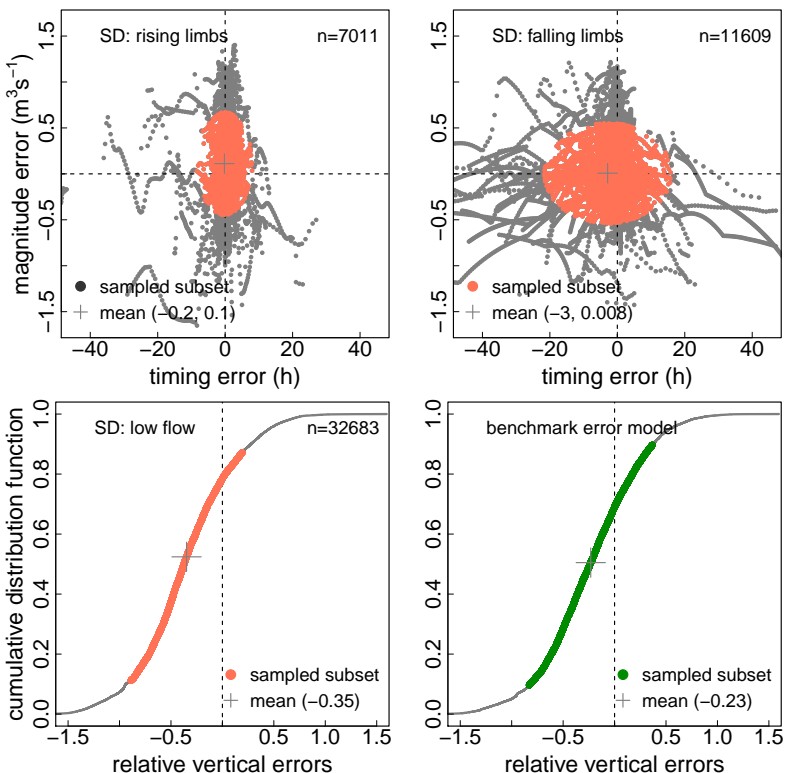

**Figure 5.** One and two-dimensional error distributions from the case study. The upper row contains Series Distance $(SD)$ results for the rising and falling limbs. The left panel in the lower row shows the one-dimensional $SD$ distribution of errors for the periods of low flow. The panel in the bottom right contains the 1-d distribution of magnitude errors of the benchmark model $(BM)$ for the entire time series. The highlighted subset represents the 80 % subset used to construct the uncertainty envelopes. Distribution statistics are provided in Table 1. To improve the readability of the upper two panels we restricted their timing-axes to the range $[-45\ 45]$. The number of outliers, points outside the range mean$\pm$3 standard deviations $([-42\ 36])$ was < 1% for the falling limbs and one order of magnitude less for the rising limbs. The dotted lines highlight the origins (all panels).

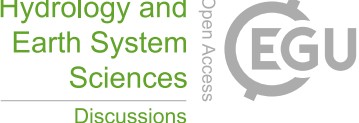



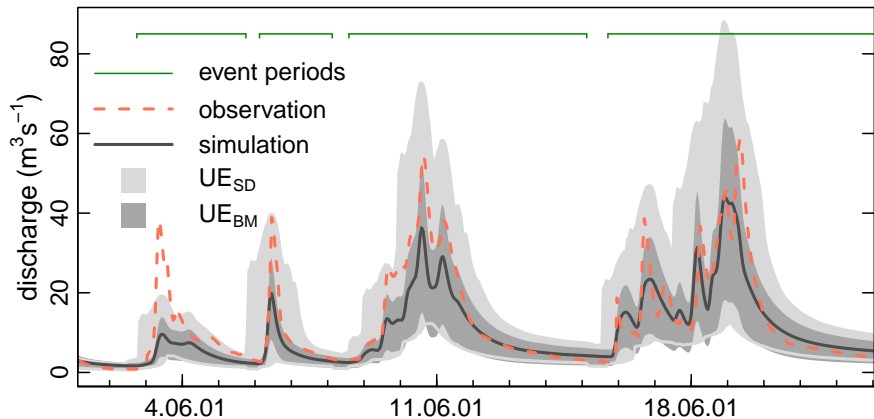

**Figure 6.** Time series detail showing the resulting 1- and 2-dimensional uncertainty envelopes around a historic streamflow simulation. The envelopes were derived upon Series Distance ($UE_{SD}$) and the benchmark approach ($UE_{BM}$) respectively, using error dressing.





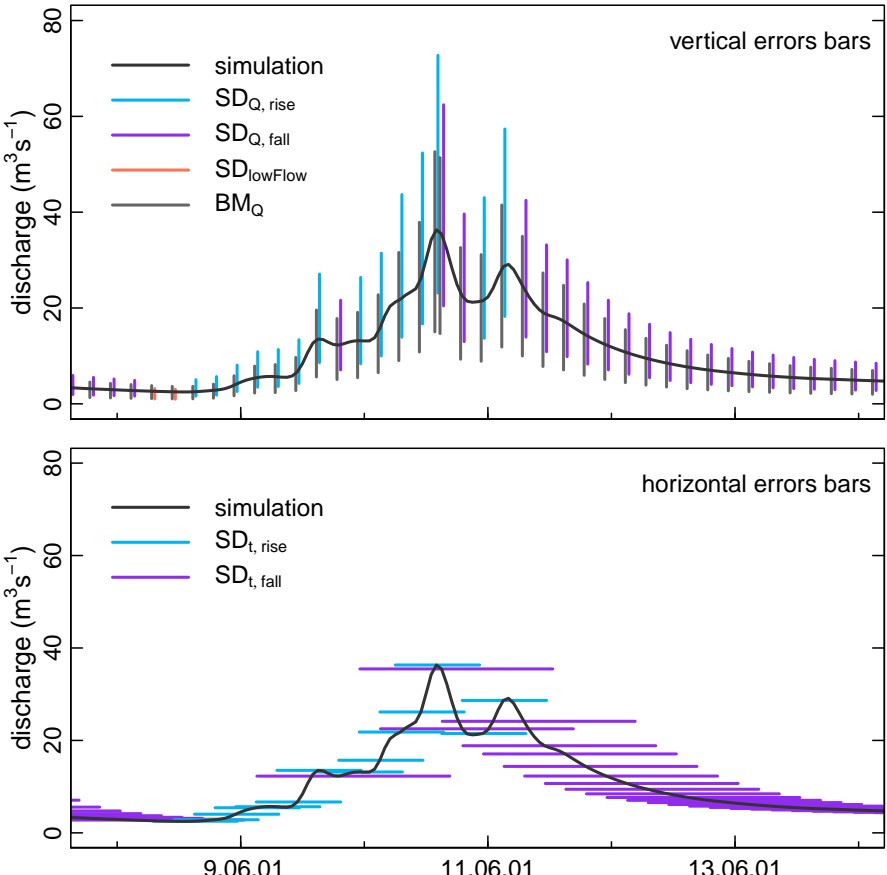

**Figure 7.** Vertical and horizontal error bars. The upper panel shows magnitude error bars ($Q$) for the Series Distance ($SD$) method and the benchmark ($BM$) approach. For $SD$ different error bars are drawn for low-flow conditions, rising (rise) and falling (fall) limbs. In the $BM$ case the same error bars are applied in all cases. The lower panel shows the corresponding timing error bars ($t$) of $SD$ (not available for $BM$), again separately for the rising and falling limbs. To improve readability we plotted error bars only every third hour and introduced a slight time offset between $SD$ and $BM$ (upper panel only). Both panels show a subset of the hydrograph section depicted in Fig. 6 and rest on the same data.





**Table 1.** Statistical properties of the individual Series Distance ($SD$) and benchmark ($BM$) error distributions from the case study. For the entire distribution we provide the first and third quartile, the mean, median and the percentage of outliers (data points which are more than three standard deviations apart from the mean). For the subset we provide the sampled upper (maximum) and lower (minimum) boundaries. The $SD$ subscripts refer to errors in magnitude ($Q$) and timing ($t$) separately for the rising ($rise$) and falling ($fall$) limbs, respectively. $SD_{LF}$ provides results for the periods of low flow.

| Error distribution | entire distribution | | | | | sampled subset | |
|---|---|---|---|---|---|---|---|
| | 25%-quartile | mean | median | 75%-quartile | %-outlier | minimum | maximum |
| $SD_{Q,rise}\ (-)$ | -0.15 | 0.11 | 0.13 | 0.39 | 0.7 | -0.44 | 0.67 |
| $SD_{Q,fall}\ (-)$ | -0.23 | 0.01 | 0.01 | 0.25 | 0.5 | -0.54 | 0.55 |
| $SD_{t,rise}\ (h)$ | -0.50 | -0.22 | 0.66 | 1.60 | 2.1 | -8.41 | 7.98 |
| $SD_{t,fall}\ (h)$ | -3.89 | -2.87 | 0 | 1.56 | 2.9 | -21.61 | 15.86 |
| $SD_{LF}\ (-)$ | -0.64 | -0.35 | -0.37 | -0.06 | 0.1 | -0.89 | 0.19 |
| $BM\ (-)$ | -0.54 | -0.23 | -0.24 | 0.09 | 0.1 | -0.83 | 0.37 |




**Table 2.** Coverage ($\phi$), precision ($PRC$) and relative precision ($PRC^*$) of uncertainty envelopes. $UE_{SD}$ and $UE_{BM}$ denote Series Distance and benchmark error model, respectively. The last column ($p$) provides the percentage of sampled values of the corresponding distribution(s).

| Uncertainty envelope | $\phi\,(-)$ | $PRC\,(m^3\,s^{-1})$ | $PRC^*\,(-)$ | $p\,(\%)$ |
|---|---|---|---|---|
| $UE_{SD}$ | 80.5 | 8.2 | 1.3 | 76 |
| $UE_{BM}$ | 80.0 | 5.1 | 1.0 | 80 |