# Peer review of "Disentangling timing and amplitude errors in streamflow simulations"

_Hydrology and Earth System Sciences, 2016_

## Referee Comment (RC1) · Anonymous Referee #1 · 9 May 2016

Overall, I like this paper very much.

But I am a little disappointed: the metric proposed here is not new. The authors already published it. They have a new improved version, this is great: but in this case, you should show us on some "large enough" dataset that the new version of the metric is better than the previous version.

Naturally, this is not easy: on which basis would you show us that the new version is "better"? The fact that we have two different numerical values proves nothing of course. You could do again what you have done with the survey of experts in Crochemore et al. (2015): is one of the versions better correlated with the opinions of the experts? An alternative would be to use the new metric to calibrate and multiple metrics to validate. If you can get better numerical results in validation on a range of numerical criteria by

"extracting" information from data during the calibration process with the new version, this would indicate a "better" criterion.

---

## Author Comment (AC1) · 11 May 2016

We thank the anonymous reviewer for his/her constructive comments. The main point of critique is that we have not provided proof that the new version of Series Distance (SD) outperforms the previous version as published in Ehret and Zehe (2011). This perceived shortcoming is most likely based on a misunderstanding which we will try to clarify in the following: The core improvements of SD as presented in the new manuscript are extensions or examples for applications rather than changes of the core idea. It is hence difficult to compare the old and the new version in a meaningful way. There are however two novel aspects in the paper:

The first is the development of a fully automated 'coarse-graining' scheme which effi-ciently mimics the techniques applied in visual hydrograph inspection by humans. This

method can be used for an automated application for the SD which was not the case for the previous version. The use of a fixed no-event threshold is over simplistic and proved to be cumbersome in practical applications. The coarse-graining procedure can also be applied on its own right to derive time series statistics such as those provided in the first paragraph of section 5.3. Both examples expand the functionality of the SD concept, while the core concept remains unchanged.

The second is an application of the SD concept for an efficient and intuitive evaluation of the importance of timing errors in continuous streamflow simulation. Here we provide a method to construct 2-dimensional uncertainty envelopes based on the 'error dressing' concept, which was not present in the 2011 paper. While the error dressing method for 1-dimensional applications is not new, it has in the new manuscript been further developed for application to 2-dimensional error distributions and their visualization (see Figures 6 and 7).

We hope that we could clarify that the main purpose of the new manuscript is not to replace the 2011 SD core concept with a new one, but to report on useful additions and potential applications. In order to give the reader an idea of SD performance, we also have provided a comparison with a benchmark model.

We will consider this point in a revised version of the manuscript; If questions remain we will be happy to answer them.

---

## Referee Comment (RC2) · Anonymous Referee #2 · 13 May 2016

This paper is about recent developments in the Series Distance (SD) approach to analysing simulated streamflow hydrographs. SD is one of a small group of approaches that automate processes similar to those performed when a hydrologist visually compares a simulated hydrograph against the corresponding observed hydrograph. There is independent evidence that such approaches have merit and could be useful in operational rainfall-runoff modelling (Lines 771-4), so new work on SD is to be welcomed. The SD approach is quite elaborate, but software has been made available (Line 777).

In terms of scientific novelty, the interest lies in the coarse graining algorithm, which is an optimisation procedure designed to find the best way to break the hydrographs into segments, such that segment 1 in the simulated hydrograph is matched with segment 1 in the observed hydrograph, segment 2 with segment 2, etc. This is similar to the unconscious process when a hydrologist visually links features (e.g. rising limbs,

short-term rainfall responses, and recessions) in a simulated hydrograph to the corresponding features in the observed hydrograph. Coarse graining is therefore a central part of what could be called the pattern matching procedure in SD; pattern matching is of fundamental interest and importance in all work based on the "visual" analysis of hydrographs. The secondary part of the pattern matching in SD, the matching at fine scale, uses linear interpolation (Line 270). My interpretation of this is as follows. Say, for example, that coarse graining gives that a simulated segment starting at time $t_s$ and lasting for $T_s$ hours corresponds to the segment $t_o, T_o$ in the observed hydrograph. Using linear interpolation, the timing error, $e$, associated with the points at fraction $x$ along these segments is:

$$e(x) = t_o - t_s + x(T_o - T_s) \text{ Equation R1}$$

The amplitude error associated with this is the difference between the simulated discharge at time $t_s + xT_s$ and the observed discharge at time $t_o + xT_o$.

There is simply too little discussion, exploration and testing of the coarse graining algorithm in the manuscript. For example, the final term in the objective function is for amplitude errors, so would seem to have central importance, but this term is switched off in the single example presented in the manuscript (line 470-1). It was switched off on the grounds that this was sufficient in a "proof-of-concept" study (line 474). For this work to be scientifically sound, the coarse graining algorithm needs to be explored fully for several types of hydrograph, and tested properly, in detail, against an appropriate benchmark. HMA (Line 552) might be suitable as a benchmark, especially given that it is a very simple algorithm.

A large part of the manuscript is about "error dressing". This is a statistical method used to obtain uncertainty clouds for simulated hydrographs (i.e. clouds of points that show where the actual hydrograph might lie). Error dressing involves: (1) collecting together the timing and amplitude errors into pools (i.e. the errors detected using coarse graining and linear interpolation); (2) drawing from these pools; and (3) applying

the draws to the simulated discharge to generate an uncertainty cloud. I found the work on error dressing unconvincing (see below). Perhaps it should be removed from the manuscript to make more room for demonstrations and discussions on coarse graining.

The few results shown for error dressing (e.g. Figure 6) show that it introduces considerable false inflation (so gives over-large error clouds), which makes the clouds difficult to interpret physically and limits their usefulness operationally. False inflation is a sign that the link has been lost between the actual local errors and the errors drawn to represent the local errors. This is not surprising given that the draws are from pools that are collected from the hydrographs as a whole, so are relevant to a huge range of different types of discharge response, and not just to the response at the time when the draw is applied. There is a nod to refining the pools, by the use of separate pools for rising and falling segments, but this, clearly, is not enough to avoid substantial inflation. Note that the use of linear interpolation in the fine-scale pattern matching adds to inflation because it neglects local information about timing. Rather than depending on local (i.e. within segment) timing, Eq. R1 shows that the local timing errors are assumed to depend only on the t and T values generated in coarse graining.

Other Points

(Line 25) The word "elaborated" does not work here.

(Line 241) Is some normalising factor or term needed here to make ISEG sum to unity?

(Line 245) This process seems to reduce the number of segments by two. What happens if an odd number needs to be eliminated.

(Line 254) It is not entirely clear why the name "coarse graining" was chosen, especially given that this required citing two otherwise irrelevant papers about other things that are commonly called coarse graining.

(Line 330) What is done if there is no local minimum in the objective function?

(Line 454) It is the gold standard in work like this to use split-sample testing because

this is the best way to test how the method would work when used operationally. Split-sample testing is trivial to apply, so there seems no reason not to use it here. The defence that the example is used simply to aid "discussion of the SD concept" (line 453) is very weak.

(Line 561) An advantage is claimed for SD that "unique relationships of points in obs and sims are established". This advantage, however, comes from using linear interpolation, which, as discussed earlier, comes at the cost of neglecting local (within segment) timing information.

---

## Referee Comment (RC3) · Anonymous Referee #3 · 16 May 2016

Summary

This study focuses on the Series Distance criterion (SD) introduced in Ehret and Zehe (2011) to consider both magnitude and timing errors when evaluating streamflow simulations. The paper first details the principle of the method, its technical aspects, as well as the improvements brought to SD since the work of Ehret and Zehe (2011). It then proposes to apply SD to represent the uncertainties of deterministic simulations. The authors use a case study to illustrate and evaluate the ensemble dressing approach based on SD and based on a benchmark approach.

General comments

I really enjoy this article. I think it is well-structured and I found it pleasant to read because the authors set the context for each step and well describe the concepts. The

illustrations well support the explanations of SD.

Overall, I would advise shortening Sections 2.2 and 2.3 and adding some text to better explain and discuss the new aspects of SD that are introduced in this paper. In Section 3, I found very interesting the use of the proposed error to build uncertainty ranges. However, in Section 5.3, I don't understand why the uncertainty ranges are so different even though they were both built to achieve an 80 % coverage. My questions on this section are listed in the detailed comments.

Detailed comments

Lines 151-162 : transform this list into a table so that the reader better visualizes the new aspects proposed. Here, the "New" sections are lost at the end of the sentences

Line 176 : I think "low-flow" should be replaced with "low flows" when it is not an adjective.

Line 200 : "this approach has been shown to work well" : could you specify ?

Lines 262-263 : I recommend removing "(or vice versa, if a falling segment is dissolved)". The sentence line 261 follows the example on line 249, and I don't think the information in parenthesis necessary to understand as it is already clear. More generally, I would recommend removing most parenthesis in the article and either include the content in the text or remove it. This could further improve the reading of the article.

Lines 268-271: "Since the segments can differ in length...between the time series edge nodes." Here, you say that you interpolate between edge nodes to obtain the same number of points to compare between the obs and sim, and, in the case of the obs time series, each edge node corresponds to an actual observation. Is that right? If so, my question is: if you linearly interpolate a segment between edge nodes in the obs time series, won't you create "fictive" observations by filling a segment? And wouldn't this "falsify" the evaluation of your simulation time series? Could you clarify this?

Line 290: "In this case," a space is missing after the comma.

Line 317: "obs an sim", "an" misses the final "d".

Line 367: Shouldn't the uncertainty ranges capture a significant portion of "observed" values rather than "simulated" values?

Sections 3.1 and 3.2: I am not sure I understand how you use the relative contribution to the total error to select your sample. For instance, in the one-dimensional case, do you assume that your uncertainty range is symmetrical around your simulated value? Since the relative contribution is positive, do you pick the value v that corresponds to 80% of the error and use it to build both the lower (-v) and upper range (+v) ? If not, the upper left graph of Figure 4 may have misled me. Or do you directly use the error values behind the relative errors within 0 and 80% and select the largest negative one to build the lower range and the largest positive one to build the upper range?

Line 398: Don't all $ds^2$ sum up to 100?

Line 449: Remove one of the "period".

Lines 552 and 773: The reference for Ewen is not formatted properly.

Line 661: I am not sure about the use of "relevant" here.

Lines 670-672: Just a question, in your opinion, could this be addressed by applying the dressing to adapted time steps? E.g. so that the distance between two dressed time steps is equal to the error in timing? Would there be an optimal time step to apply the dressing (one for which the percentage of sampled errors and the overall coverage would match)?

Section 5.3.2: (1) From Figure 6, it is not intuitive that the SD and BM approaches have the same coverage. Is it specific to this example? Do you have cases when there are more observations outside the SD envelope than outside the BM envelope that would compensate for the case you show? (2) Based on how the SD envelope is constructed, you would expect a third of the 20 % outside the envelope occur in low flows, a third in rising limbs and a third in falling limbs. On the opposite, the 20 % falling outside the BM

envelope can occur whenever in the time series, and can, for instance, always occur when it is harder to model streamflow, i.e. during events. Does this have an impact here? How does that affect the results?

Lines 700-702: From this sentence, it may seem that low flows also have a 2-d error distribution, but errors in low flows are 1-d. Could you clarify this?

Lines 719-720: What would be the difference between "uniting" and "intersecting" in this case?

Line 723: "as proposed by" I believe a reference is missing here.

Figure 5: The dot of the sampled subset in the upper left graph is black whereas the sampled subset is orange.

---

## Author Comment (AC2) · 14 Jun 2016

**Reply to Referee #2**

We sincerely thank Referee # 2 for the thorough review and for providing excellent suggestions. Our manuscript will considerably benefit from it.

RC#2 elaborates on two general aspects:

A) Coarse-graining, i.e. our pattern matching procedure which breaks the simulated and the observed hydrographs into segments (step 1) and conducts the matching and comparison of the two time series using "fine scale linear interpolation" as RC#2 put it (step 2).

RC#2 correctly describes the calculation of the error in SD using the Equation R1.

The major point of criticism raised by RC#2 with respect to coarse-graining is that "there is simply too little discussion, exploration and testing of the coarse graining algorithm in the manuscript (...) to be scientifically sound, the coarse graining algorithm needs to be explored fully for several types of hydrograph, and tested properly, in detail, against an appropriate benchmark".
Later in the review RC#2 provides an important additional note which is "that the use of linear interpolation in the fine-scale pattern matching adds to inflation because it neglects local information about timing. Rather than depending on local (i.e. within segment) timing, Eq. R1 shows that the local timing errors are assumed to depend only on the t and T values generated in coarse graining."

Our reply to this major point is:
1) We do agree that the coarse graining algorithm has not been fully explored yet. The analysis and evaluation of coarse graining in quantitative terms is however very difficult as the breaking of the hydrographs into segments is subjective and thus, to a certain degree arbitrary. Since there is no reference for the breaking of the hydrograph available it is difficult to meaningfully compare coarse-graining to a benchmark approach. Essentially, only a visually comparison seems meaningful to us. This would however require many plots of different hydrographs and flow regimes which is difficult to realize within in a paper or supplement. For this reason we decided to provide software such that any interested reader can find out for him/herself whether the proposed method suits his or her needs.
2) RC#2 is correct that the accuracy of the SD procedure crucially depends on t and T, and thus, on the accuracy of the coarse-graining. Poor coarse-graining yields incorrect matched segment pairs (compare for instance those in the upper panels of Fig. 3). These inevitably introduce large timing and possibly even large magnitude errors and may thus also add to false inflation. For this reason we clearly state that coarse-graining is of fundamental importance for SD (line 542): "...the quality of the segment matching largely determines the quality of the subsequent matching of obs and sim points and hence the quality of the SD error calculation".

Due to the large importance of coarse-graining we propose to illustrate the sensitivity of the weighting factors in the objective function (Eq. 2) in more detail in the revised version of the manuscript.

B) The second major point of criticism refers to the "error dressing" concept which is regarded unconvincing by RC#2. As reason RC#2 states that only "few results for error dressing are shown and that the method introduces considerable false inflation (overly large error clouds), which makes the clouds difficult to interpret physically and limits their usefulness operationally." RC#2 also raises the

questions whether the entire section on error dressing should be removed from the manuscript to make more room for demonstrations and discussions on coarse graining.

Our answers to this major point is:

1) We consider the joint visualization of timing and magnitude errors a fundamental and important part of the manuscript as this is rarely done in hydrological papers.

2) The reasons for false inflation are manifold and cannot be attributed to error dressing alone. Possible causes for false inflation include: i) poor coarse-graining results as these will yield large timing errors, ii) poor definition of the error pools where the samples are taken from in error dressing, iii) the use of an absolute timing error model (Eq. 6) instead of a relative one like for the magnitude errors and iv) the neglecting of the local, within segment, auto-correlation of timing errors. Each of these sources can cause large timing errors and thus, contribute to false inflation.

For these reasons we would prefer to keep the section on error dressing as it is. We provided it as one possible application of SD although there are of course more elaborate methods available. We propose however to discuss the reasons for false inflation in more detail and to specify possible ways forward.

Other points
* * *
RC#2: (Line 25) The word "elaborated" does not work her

AR: We don't see the problem of the use of "elaborated" in this context. Could you please specify the reason why it does not work. Change to "detailed", "sophisticated", "ambitious",???

RC#2: (Line 241) Is some normalizing factor or term needed here to make ISEG sum to unity

AR: ISEG sums to unity as it is calculated based upon the relative duration ($dt*(i)$), and the relative magnitude change ($dQ*(i)$). We obtain $dt*(i)$ and $dQ*(i)$ by normalizing $dt(i)$ by the total duration and $dQ(i)$ by the sum of the absolute magnitude changes of the entire event. Equation 1 does hence essentially sum up to unity.

RC#2: (Line 245) This process seems to reduce the number of segments by two. What happens if an odd number needs to be eliminated?

AR: RC#2 is correct, each step of aggregation reduces the number of segments by two. It is however not possible that an odd number of segments needs to be eliminated. During the pre-processing we trim the time series, if required, to ensure that both hydrographs do start and end with an identical segment type, i.e. either a rising or falling segment.

RC#2: (Line 254) It is not entirely clear why the name "coarse graining" was chosen, especially given that this required citing two otherwise irrelevant papers about other things that are commonly called coarse graining.

AR: Good point. We spent a lot of time thinking about a name that has least overlap with existing procedures and found this to be the best. However, we like the suggestion by the referee and will change the wording to 'pattern matching procedure (PMP)' in a revised version of the manuscript.

RC#2: (Line 330) What is done if there is no local minimum in the objective function

AR: The minimum is taken in every case independent if it is a local minimum or not. Local minima occur in the coarse graining of "complex" multi-peak events with large number of coarse graining steps. In "simple" events where no or little coarse graining is required the objective function values often increase fairly linear. We will add a brief comment on this issue at line 330.

RC#2: (Line 454) It is the gold standard in work like this to use split-sample testing because this is the best way to test how the method would work when used operationally. Split-sample testing is trivial to apply, so there seems no reason not to use it here. The defence that the example is used simply to aid "discussion of the SD concept" (line 453) is very weak.

AR: We agree that split-sampling is widely used in this context. However, we do not see the benefit in providing additional information on the method by means of using split-sampling here. Again, the main purpose of our case study is to illustrate the method using real-world data and not whether the achieved coverage differs by some percentage between "calibration" and "validation".

RC#2: (Line 561) An advantage is claimed for SD that "unique relationships of points in obs and sims are established". This advantage, however, comes from using linear interpolation, which, as discussed earlier, comes at the cost of neglecting local (within segment) timing information.

AR: In our perception it is important to perceive the temporal order of both the simulated and the observed values. For this reason we uniquely compare the first point in obs to the first point in sim, the second to the second, etc. (compare lines 206-211). It is this assumption which justifies the use of linear interpolation,  not vice versa.

---

## Author Comment (AC3) · 14 Jun 2016

**Reply to Referee #3**

We thank the anonymous Referee #3 for the positive feedback and for providing so many detailed comments. The latter will help us to improve the quality of the manuscript. Here we provide a point-by-point list of author replies (AR) to all issues raised by the reviewer.

**General comments**
* * *
RC#3: Overall, I would advise shortening Sections 2.2 and 2.3 and adding some text to better explain and discuss the new aspects of SD that are introduced in this paper. In Section 3, I found very interesting the use of the proposed error to build uncertainty ranges. However, in Section 5.3, I don't understand why the uncertainty ranges are so different even though they were both built to achieve an 80 % coverage. My questions on this section are listed in the detailed comments.

AR: The sections 2.2 and 2.3 actually describe all new aspects of SD. For this reason we are not sure about where to shorten these sections. Could you please clarify? Our answers concerning the questions about section 5.3 are provided in the detailed comments.

**Detailed comments**
* * *
RC#3: Lines 151-162: transform this list into a table so that the reader better visualizes the new aspects proposed.
AR: Thank you for this suggestion. We will clarify the novel aspects and compare them to the previous version.

RC#3: Line 176: I think "low-flow" should be replaced with "low flows" when it is not an adjective.
AR: We will do that

RC#3: Line 200 : "this approach has been shown to work well" : could you specify ?
AR: This is difficult to show in a simple graph or statistic. Rather, this is the authors' perception after applying the SD approach to many different discharge time series in both the 'many events' and 'single, long event' mode. 'Shown to work well' in this context means even in the 'single, long event' mode, SD linked parts of obs and sim time series that visually appeared to be matching segments within matching events. As we provide the SD code and test data together with the article, this can easily be tested by interested users. We will include these two points in a revised version of the manuscript.

RC#3: Lines 262-263 : I recommend removing "(or vice versa, if a falling segment is dissolved)". The sentence line 261 follows the example on line 249, and I don't think the information in parenthesis necessary to understand as it is already clear. More generally, I would recommend removing most parenthesis in the article and either include the content in the text or remove it. This could further improve the reading of the article.
AR: Thank you for pointing this out. We will revise the article again with respect to the wording and remove words in parenthesis wherever possible.

RC#3: Lines 268-271: "Since the segments can differ in length. . .between the time series edge nodes." Here, you say that you interpolate between edge nodes to obtain the same number of points to compare between the obs and sim, and, in the case of the obs time series, each edge node

corresponds to an actual observation. Is that right? If so, my question is: if you linearly interpolate a segment between edge nodes in the obs time series, won't you create "fictive" observations by filling a segment? And wouldn't this "falsify" the evaluation of your simulation time series? Could you clarify this?

AR: You did understand right. Due to the linear interpolation we obtain "fictive" observations. We do however not understand why this should falsify the evaluation as the same procedure is applied to the simulated time series. The idea of the SD method is to identify and to compare points which are "hydrologically similar". This essentially requires to compare points which do not share the same abscissa and which do not coincide with the hourly observations.

RC#3: Line 290: "In this case," a space is missing after the comma.

AR: we will add the space.

RC#3: Line 317: "obs an sim", "an" misses the final "d".

AR: we will add the missing "d"

RC#3: Line 367: Shouldn't the uncertainty ranges capture a significant portion of "observed" values rather than "simulated" values?

AR: This is of course correct. We will change it in the revised version.

RC#3: Sections 3.1 and 3.2: I am not sure I understand how you use the relative contribution to the total error to select your sample. For instance, in the one-dimensional case, do you assume that your uncertainty range is symmetrical around your simulated value? Since the relative contribution is positive, do you pick the value v that corresponds to 80% of the error and use it to build both the lower (-v) and upper range (+v) ? If not, the upper left graph of Figure 4 may have misled me. Or do you directly use the error values behind the relative errors within 0 and 80% and select the largest negative one to build the lower range and the largest positive one to build the upper range?

AR: The second case is correct. We directly use the error values behind the relative errors within 0 and 80% and select the largest negative one to build the lower range and the largest positive one to build the upper range. We do hence not assume that the uncertainty range is symmetrical around the simulated value. Please compare also the results from the case study (fig. 5) where this is the case. We do however agree that the sketch in Fig. 4 is a little bit misleading. There reason for this is that the sketch in Fig. 4 is, for the sake of simple presentation, based on "normally distributed" random numbers which do of course yield symmetrically centered envelopes.

RC#3: Line 398: Don't all ds2 sum up to 100?

AR: This is correct, we will change it.

RC#3: Line 449: Remove one of the "period"

AR: Of course.

RC#3: Lines 552 and 773: The reference for Ewen is not formatted properly

AR: You are right, we will correct the formating.

RC#3: Line 661: I am not sure about the use of "relevant" here.

AR: We agree. Maybe "important" fits better. We will change it.

RC#3: Lines 670-672: Just a question, in your opinion, could this be addressed by applying the dressing to adapted time steps? E.g. so that the distance between two dressed time steps is equal to

the error in timing? Would there be an optimal time step to apply the dressing (one for which the percentage of sampled errors and the overall coverage would match)?

AR: We are not sure if we correctly understand your question. We are however grateful for asking it as this points needs to be clarified. We expect that there are different reasons for this problem: The most important ones are probably in the selection of the error model and in the definition of the error distributions. In the manuscript we applied SD using a "relative" magnitude error model but an "absolute" horizontal error model. The result is that the same (static) timing error is applied to all time steps. One possible way out would be to formulate the timing error in a relative way, e.g. using average event duration. This would very likely narrow the timing errors. Another alternative would be to further differentiate the error distributions according to streamflow magnitude since we do not expect that the same timing errors do occur at each flow rate. Last, the current version of SD does not account for the auto-correlation of the errors which is typically high in streamflow data. Considering it in both, the calculation and in the application of the errors would very likely narrow the uncertainty envelopes to a significant degree. We will note on these aspects in the revised version of the manuscript.

RC#3: Section 5.3.2: (1) From Figure 6, it is not intuitive that the SD and BM approaches have the same coverage. Is it specific to this example? Do you have cases when there are more observations outside the SD envelope than outside the BM envelope that would compensate for the case you show? (2) Based on how the SD envelope is constructed, you would expect a third of the 20 % outside the envelope occur in low flows, a third in rising limbs and a third in falling limbs. On the opposite, the 20 % falling outside the BM envelope can occur whenever in the time series, and can, for instance, always occur when it is harder to model streamflow, i.e. during events. Does this have an impact here? How does that affect the results?

AR: Good point. The coverages of SD and BM shown in the plot are indeed different. The reason is that the plot contains only a subset of the entire time series: While for the entire time series the overall coverages are indeed equal for SD and BM, the values can deviate for subsets of the time series. For other subsets such as very small events classified as low flow the effect is opposite: Here the 1-D error distributions are applied to the simulation for both SD and BM. These differ however in their extent (compare the lower two panels of Figure 5), which causes that BM has a "higher" upper envelope than SD during low flow. We agree that the case depicted in Fig. 6 is not intuitive with respect to coverage and we will add an explanation to clarify this in the revised version of the manuscript.

RC#3: Lines 700-702: From this sentence, it may seem that low flows also have a 2-d error distribution, but errors in low flows are 1-d. Could you clarify this?

AR: Of course. Thank you for pointing this out. We will change the manuscript accordingly.

RC#3: Lines 719-720: What would be the difference between "uniting" and "intersecting" in this case?

AR: These terms refer to set theory. Intersecting means to use only "elements common to both error components" whereas uniting would mean to use all elements, which is what we have done at the moment. We will clarify this in a revised version of the manuscript.

RC#3: Line 723: "as proposed by" I believe a reference is missing here.

AR: This is a wording error. We will correct it.

RC#3: Figure 5: The dot of the sampled subset in the upper left graph is black whereas the sampled subset is orange.

AR: You are right, we will correct this